# Mitochondrial Dysfunction and Permeability Transition in Neonatal Brain and Lung Injuries

**DOI:** 10.3390/cells10030569

**Published:** 2021-03-05

**Authors:** Vadim S. Ten, Anna A. Stepanova, Veniamin Ratner, Maria Neginskaya, Zoya Niatsetskaya, Sergey Sosunov, Anatoly Starkov

**Affiliations:** 1Departments of Pediatrics, Columbia University Irving Medical Center, New York, NY 10032, USA; aas2337@cumc.columbia.edu (A.A.S.); zn2120@cumc.columbia.edu (Z.N.); sas2110@cumc.columbia.edu (S.S.); 2Department of Pediatrics, Icahn Mount Sinai School of Medicine, New York, NY 10029, USA; Veniamin.Ratner@mssm.edu; 3Department Molecular Pathobiology, New York University School of Dentistry, New York, NY 10010, USA; mn2452@nyu.edu; 4Neuroscience Department, Weil-Cornell School of Medicine, New York, NY 10065, USA; ans2024@med.cornell.edu

**Keywords:** prematurity, mitochondria, proton leak, lungs, brain

## Abstract

This review discusses the potential mechanistic role of abnormally elevated mitochondrial proton leak and mitochondrial bioenergetic dysfunction in the pathogenesis of neonatal brain and lung injuries associated with premature birth. Providing supporting evidence, we hypothesized that mitochondrial dysfunction contributes to postnatal alveolar developmental arrest in bronchopulmonary dysplasia (BPD) and cerebral myelination failure in diffuse white matter injury (WMI). This review also analyzes data on mitochondrial dysfunction triggered by activation of mitochondrial permeability transition pore(s) (mPTP) during the evolution of perinatal hypoxic-ischemic encephalopathy. While the still cryptic molecular identity of mPTP continues to be a subject for extensive basic science research efforts, the translational significance of mitochondrial proton leak received less scientific attention, especially in diseases of the developing organs. This review is focused on the potential mechanistic relevance of mPTP and mitochondrial dysfunction to neonatal diseases driven by developmental failure of organ maturation or by acute ischemia-reperfusion insult during development.

## 1. Introduction

Mitochondria have been increasingly recognized as organelles with various roles in guarding cellular fate in human diseases, ranging from aging-related chronic degenerative pathologies to acute tissue damage following the cell-lethal insult. However, in neonatal diseases, like white matter injury (WMI), hypoxic-ischemic (HI) brain injury, chronic lung disease/bronchopulmonary dysplasia (BPD), the potential pathogenic role of mitochondrial dysfunction is not yet appreciated and remains poorly understood.

Considering cellular fate, the majority of diseases in newborn infants could be divided into two main categories: (1) diseases caused by extensive cellular death, leading to a loss of functional tissue and organ failure, and (2) diseases driven by the organ developmental arrest rather than tissue loss, leading to organ functional immaturity and clinical signs of organ failure. While both processes, cellular death and cellular maturation, are closely interconnected, clear recognition of the leading mechanisms driving organ malfunction is critical for therapeutic success. For example, neonatal BPD or diffuse WMI in prematurely born infants are currently viewed as diseases caused by an arrested alveolar development in the lungs or by a failure of oligodendrocyte maturation into myelin-producing cells in the brain. Therefore, therapeutic strategies for the prevention and treatment of these conditions should address developmental failure mechanisms rather than the mechanisms of cellular demise. In contrast, therapeutic strategy against hypoxic-ischemic encephalopathy (HIE) requires a clear understanding of the ischemia-reperfusion (IR)-driven mechanisms of cellular death, which defines the extent of reperfusion injury to the developing brain. Thus, depending on cellular fate, death, or differentiation failure of viable cells, the mechanisms and the degree of mitochondrial dysfunction contributing to the disease evolution will differ.

### 1.1. Mitochondria and Perinatal Hypoxia-Ischemia Brain Injury

Perinatal HI-brain injury usually occurs due to acute cessation of systemic circulation at or near birth or during the immediate neonatal period. If not interrupted, HI-insult is lethal. If systemic circulation is restored promptly, then full or partial recovery of the brain and other organs is expected. The extent of this recovery will determine the absence or presence and the severity of neurological deficit. Thus, the return of the systemic and cerebral circulation (reperfusion) defines the disease state following HI insult. As the brain is susceptible to ischemia, neurological outcome following neonatal HI insult remains a primary clinical concern.

Neonatal outcomes of c-section deliveries associated with maternal cardiovascular crisis demonstrated that, when delivered within 5 min of maternal circulatory arrest, 70% of infants were free of neurological sequelae. However, if the delivery was delayed by 6–10 min, only 13% of infants developed normally [1]. Models of perinatal asphyxia and cerebral ischemia-reperfusion demonstrated that the length of ischemia is the most critical determinant of injury severity [2,3,4]. The quality of reperfusion also has a significant impact on the extent of cerebral damage. Even brief ischemic insults followed by suboptimal oxygen redelivery upon reperfusion may cause extensive cerebral tissue damage [5].

On the other hand, when a neonatal ischemic event extends over 5–7 min at a physiological temperature [6,7], reperfusion both ensures cerebral recovery and contributes to injury via the mechanisms depending on the reintroduction of oxygen and energy (e.g., oxidative stress and apoptosis). Thus, two interconnected biological processes, ischemia and reperfusion, contribute to perinatal HI brain injury. A pathophysiological key of ischemia is primary energy failure driven by acute oxygen and substrates deprivation. Pathophysiology of reperfusion is more complex, as reperfusion initiates both cellular recovery and detrimental to cellular recovery mechanisms.

### 1.2. Primary Energy Failure in Perinatal HI Insult

Primary cerebral energy failure signifies acute depletion of high-energy phosphates, adenosine triphosphate (ATP) and phosphocreatine (PCr) in the brain. Ischemia ceases electron flux in the mitochondrial respiratory chain, and mitochondrial complexes cannot maintain a proton gradient across the inner mitochondrial membrane, rendering phosphorylation of ADP to ATP impossible. Once circulation has failed, a bioenergetic crisis develops rapidly: significant depletion of PCr, ATP, and elevation of ADP, AMP occurs in 10 s, and profound depletion of energy charges occurs within 5–7 min of an acute ischemic insult [5]. For a short time, hydrolysis of residual ATP stores and the use of PCr for phosphorylation of ADP partially supports cellular energy demand, and anaerobic glycolysis becomes the primary mechanism of energy production.

In neonates, cerebral immaturity predisposes to poorer efficiency of glycolysis compared to the mature brain. Neonatal rats subjected to HI brain injury demonstrated limited activation of anaerobic glycolysis in their brains due to developmental deficiency of the glucose transporter proteins GLUT1 (glial) and GLUT3 (neuronal), which are at 7 days expressed only at ≈10% of their adult levels [8]. This suggests that in the developing brain, the acuity of primary energy failure may be greater compared to the mature brain. The immediate sequelae of the primary bioenergetic crisis is a failure of cellular structure and function-maintaining ion pumps, i.e., Na-K+ ATPase, leading to a loss of ion gradient across the cellular membrane, depolarization, and cytotoxic swelling. Experimental and clinical research offers strong evidence for the association between the severity of the primary bioenergetic crisis and poor neurological prognosis following neonatal asphyxia [9,10,11]. Besides primary energy failure, other fundamental biochemical responses to ischemia extend their contribution to injury into the reperfusion stage of the disease.

Acute oxygen deprivation results in a complete reduction of the electron-transferring components, flavins, iron-sulfur clusters, coenzyme Q and cytochromes *a*, *b* and *c*, in the mitochondrial respiratory chain. Primary Krebs cycle electron donors (NADH and FADH_2_) are also fully reduced. Ischemic over-reduction of NAD^+^ to NADH inhibits glycolytic ATP generation because oxidation of glyceraldehyde 3-phosphate requires NAD^+^. Upon reperfusion, the reintroduction of oxygen, in special circumstances, fully reduced electron carriers may leak electrons onto oxygen generating superoxide [12]. Thus, over-reduction of electron carriers during ischemia predisposes the respiratory chain to excessive electron leak and formation of reactive oxygen species (ROS) during reperfusion.

Anaerobic glycolysis produces not only ATP but also lactate. The ischemia inhibits both the pyruvate dehydrogenase complex [13] and the respiratory chain, and pyruvate cannot be used in the mitochondrial tricarboxylic acid cycle (TCA), shunting into lactate. Even in the absence of any substrate delivery, the pre-ischemic glucose and glycogen levels are sufficient to increase tissue lactate from 1.5 to 12–14 µmole/g within 2–3 min [14]. In isolated rat brain mitochondria, lactic acidosis suppresses the respiratory chain [15], potentially contributing to the ischemic bioenergetic collapse. In the developing and mature mouse brains, ischemia also results in a dramatic (≈30-fold) accumulation of succinate, while the levels of other Krebs cycle intermediates are depleted [16,17,18]. These metabolic changes have been proposed as mechanistic factors predisposing to oxidative stress upon reperfusion [18,19].

On a cellular level, ischemia causes over-excitation of neurons and oligodendrocytes due to excessive release of neurotransmitters (glutamate) and bioenergetic failure of glutamate reuptake [20]. Glutamate release and activation of the glutamate receptors, N-methyl-D-aspartate (NMDA), α-amino-3-hydroxy-5-methyl-4-isoxazolepropionic acid (AMPA receptor) and other receptors, initiate an excitotoxic cascade where downstream cellular Ca^2+^ influx plays a central role in cellular damage during reperfusion [21,22,23]. Thus, besides primary energy failure, ischemia causes biochemical changes that limit cellular bioenergetics restoration and cellular recovery upon reperfusion.

### 1.3. Reperfusion and Secondary Energy Failure in Perinatal HI Insult

The concept of secondary energy failure rests on experimental and clinical data showing the brisk, near-full, or partial restoration of high-energy phosphates in the ischemic brain upon reperfusion [10,24,25,26]. These data imply that upon oxygen and substrates redelivery, post-ischemic mitochondria are capable of ATP generation. However, following a few hours of reperfusion, 6-8 h in human neonates [27], cerebral bioenergetics progressively declined. The secondary energy failure mechanism directly relates to changes in the mitochondrial capacity to generate ATP during reperfusion.

In the Rice-Vannucci model of neonatal HI-brain injury (ligation of the common carotid artery and exposure to hypoxia), cerebral mitochondria isolated in a few minutes after the insult exhibited significantly decreased respiratory control ratio mostly due to inhibited phosphorylating respiration [28,29], the event coupled with significantly depressed complex I activity [19]. At 3 h post-reperfusion, the mitochondrial phosphorylating respiration and respiratory control ratio had partially recovered [28]. In the same mouse model, when tested at 30 min of reperfusion, post-HI mitochondria exhibited near-full restoration of complex I activity and ADP-phosphorylating capacity [19,29]. In mature gerbils, after 30 min of global brain ischemia, at 5–30 min of reperfusion, mitochondrial respiration also fully regained its capacity to phosphorylate ADP, which was significantly depressed immediately after ischemic insult. However, at 120 min of reperfusion, there was a secondary decline in mitochondrial phosphorylating capacity [30]. These reperfusion-associated changes in mitochondrial respiration were paralleled by changes in enzymatic activities of respiratory chain complexes I-III, but not complex IV [19,30]. Regardless of the extent of mitochondrial functional recovery, these and other studies [31] have shown reproducible restoration of oxidative phosphorylation after HI-insult, followed by a progressive decline of mitochondrial phosphorylating function after few hours of reperfusion.

To understand the pathogenic significance of secondary energy failure, we need to determine whether the secondary bioenergetic crisis contributes to the evolution of cellular injury or simply reflects metabolic shutdown in the dying tissue. R. Vannucci and colleagues examined temporal changes in the brain’s high-energy phosphates reserves and markers of neuronal damage during post-HI reperfusion. Based on the close association of PCr levels with the loss of neuronal protein markers and cerebral injury scores at 6–18 and 24–48 h of reperfusion, the authors proposed that secondary energy failure is the consequence rather than the cause of ultimate cellular death [26]. A study on the temporal association between neuronal protein (MAP2) loss, caspase 3 activation and secondary mitochondrial dysfunction in neonatal rats has shown the caspase 3 activation at the time-point of reperfusion, when mitochondria exhibited near-normal respiratory activity, suggesting that activation of this cell death pathway preceded secondary mitochondrial dysfunction [28]. In the model of focal ischemia-reperfusion injury in mature rats, electron microscopy of mitochondria isolates revealed a disruption of the matrix structure of mitochondria at the end of ischemia with progression to the rupture of the inner and outer mitochondrial membranes at 2 h of reperfusion. These changes were paralleled by progressively increased cyclophilin D expression in the isolated mitochondria, mitochondrial depolarization, and histopathological evidence of cellular death [32]. Combined with the above-referenced reports, these data show that mitochondrial swelling, suggesting the presence of elevated mitochondrial proton/ion leak across the inner membrane, developed in parallel with histopathological evidence of brain damage. As excessive proton leak uncouples mitochondria, these data also suggest the causal role of secondary mitochondrial dysfunction initiated by reperfusion-driven activation of mPTP in cellular damage. In the same model of 2 h of focal cerebral ischemia in rats, during reflow, the levels of high energy phosphates in the ischemic core, penumbra, and striatum recovered to values near those of control but then decreased with the continuation of reperfusion. Importantly, the possibility of rebound ischemia in this secondary energy failure was precluded by regional cerebral blood flow and glucose concentration values, which were significantly greater than in ischemia [33]. These authors concluded that the evolution of infarct during reflow results from loss of ATP and its synthesis.

Upcoming research will clarify the role of secondary energy failure in reperfusion injury of the developing brain and will answer two important questions: (a) Is mPTP responsible for mitochondrial swelling detected in reperfusion? (b) Is this mPTP a viable therapeutic target for attenuation of secondary mitochondrial dysfunction in neonatal HI brain injury? Of note, hypothermia’s neuroprotective action had been linked to a decreased cellular energy demand in asphyxiated neonates [34].

### 1.4. Mitochondrial Permeability Transition Pore(s) (mPTP) and Secondary Energy Failure

Mitochondrial matrix content is isolated from the cytosol by two membranes, the ion-impermeable inner membrane and outer mitochondrial membrane, which is permeable to ions and small molecules (like ATP) via a voltage-dependent anion-selective channel (VDAC) [35]. The import of nuclear-coded proteins to mitochondria is assisted by the translocase of the outer membrane (TOM) and by the translocase of the inner membrane (TIM). Impermeability of the inner mitochondrial membrane for ions/protons supports mitochondrial membrane potential (Ψm) [36], which serves as the proton motive force for ATP synthase activity. Under normal conditions, mitochondrial inner membrane can become permeable only during transient activation of mitochondrial ion channels [37]. However, in a broad range of pathologies, loss of mitochondrial function and cell death is linked to the loss of Ψm due to an increase in membrane permeability—usually termed as Mitochondrial Permeability Transition [38] (Figure 1). It is suggested that, depending on the severity of stress, permeability transition can be presented as the large mPTP which is cell-lethal, high-conductance mPTP, or as low-conductance mitochondrial ion/proton leak [39,40] (Figure 1). mPTP opening is mediated/regulated by the protein Cyclophilin D, which is the target for the inhibitor of mPTP Cyclosporine A (CsA) [41,42]. The exact molecular identity of the mPTP is still debated. There are two main molecular candidates that can form a large “mPTP-like” channel in the model membranes, Adenine Nucleotide Translocase (ANT) and ATP synthase (ATPase) (Figure 1). It was suggested that in vivo, these complexes could form mPTP during oxidative stress and mitochondrial calcium overload [43,44,45,46]. Functional significance of activated mPTP relates to rendering mitochondria ATP production severely impaired (Figure 1).

HI-insult in the brain results in excessive release of glutamate, activation of NMDA, and AMPA glutamate receptors. The primary detrimental mechanism, downstream to glutamate-NMDA and AMPA interaction, is excessive influx of intracellular Ca^2+^ [20,48]. Mitochondria are the organelles that actively regulate Ca^2+^ level in the cytosol by taking Ca^2+^ up into their matrix via a mitochondrial Ca^2+^ uniporter and slowly exporting Ca^2+^ by Ca^2+^/Na^+^ and H^+^/Ca^2+^ exchangers [49,50]. Mitochondrial Ca^2+^ buffering prevents cell damage caused by the toxic effects of excessive Ca^2+^. This mechanism of Ca^2+^ release via transient activation of mPTP has been reported in physiological conditions in cardiomyocytes and other cells [51,52]. Following acute ischemia in the heart, brain, liver, or kidney, a massive Ca^2+^ overload results in Ca^2+^-induced permanent opening of mPTP. Compared to transient physiological activation of the low-conductance mPTP, post-ischemic mPTP has been characterized as a permanent formation of a high-conductance channel causing mitochondrial swelling and tissue necrosis [53]. A useful framework concept for the mPTP-driven evolution of reperfusion brain injury can be presented as follows: Ischemic glutamate-receptor activation → excessive Ca^2+^ cellular/mitochondrial influx → Ca^2+^-dependent opening of permanent mPTP → mitochondrial uncoupling and swelling which eventuates in mitochondrial matrix disintegration (Figure 1). This concept explains reasonably well the mechanisms for secondary inhibition of mitochondrial ATP production, mPTP-dependent necrosis, and activation of the apoptotic cell death pathway. Opening of mPTP causing mitochondrial swelling may promote a release of cytochrome *c* and other mitochondrial apoptosis-inducing components into the cytosol. A loss of cytochrome *c* also interrupts electron transfer in the respiratory chain contributing to mitochondrial malfunction [54,55]. Thus, the concept of permanent mPTP activation considers mitochondria as the site and mitochondrial dysfunction as the event initiating necrotic and apoptotic cell death pathways in ischemia-reperfusion injury [56]. Several reports demonstrating cerebral and cardiac protection afforded by inhibition of mPTP opening following ischemia and reperfusion in mature and immature animals support a central role of mPTP in mediating cellular injury [53,57,58].

There are, however, fundamental unresolved questions. Firstly, the molecular identity of mPTP remains cryptic. Secondly, studies addressing the role of mPTP in HI neonatal brain injury are conflicting. Some authors have reported a beneficial or partially beneficial effect of blockage of Ca^2+^-dependent mPTP opening with cyclosporine A [59,60]. In contrast, mPTP inhibition with a different inhibitor (GNX-4728) in the same model failed to afford neuroprotection [61]. Cyclosporin A is a nonspecific inhibitor of the cyclophilin D, the protein which is well-characterized as a critical mediator of mPTP opening. A pathogenic contribution of the cyclophilin D-dependent mPTP opening in focal ischemic mature brain injury has been strongly supported by a significant attenuation of the infarct volumes in cyclophilin D knock-out adult mice [57]. In contrast, cyclophilin D-deficient immature mice exhibited exacerbation of the brain injury following HI-insult, yet mature cyclophilin D-deficient mice, subjected to the same model, were protected [47]. These data suggest that the effects of cyclophilin D relevant to the cellular fate following ischemic insult may be completely different depending upon the developmental stage of the brain, cell pro-survival in the immature, neonatal brain (Figure 1) and cell-lethal in the mature brain. Conflicting data with the use of mPTP inhibitors in the same species and in the same model challenge a fundamental role of mPTP in the pathophysiology of ischemia-reperfusion injury to the developing brain.

Given that the initial effect of mPTP is an elevated proton leak, it is also important to note that to date, studies on mitochondrial respiration conducted during different time-points of reperfusion did not reveal elevated resting respiration rate [28,29]. However, in adult rats at the same time of reperfusion after focal ischemic insult, electron microscopy revealed swollen mitochondrial matrix consistent with excessive proton leak [32]. Our initial patch-clamp experiments with neonatal (*p*10) mouse model of HI brain injury are in agreement with these data and demonstrated significantly elevated CsA-insensitive electrical conductance in mitoplasts isolated from the ischemic brain at 30 min of reperfusion, compared to naïve controls and mitoplast obtained at 0 min of reperfusion (our unpublished observations). It has been reported that in vitro Ca^2+^-induced mPTP exhibited resistance [62] or limited sensitivity to CsA [63] in isolated rat brain mitochondria. Thus, it is possible that the sensitivity of cerebral mPTP to CsA, and therefore the role of cyclophilin D in post-HI mPTP activation is developmentally regulated. Overall, existing data on the pathogenic role of cyclophilin D-dependent mPTP in the evolution of HI brain injury in the developing brain are conflicting not only within the field of the developing brain injury but also inconsistent with data obtained in the models of mature HI and IR brain injuries. More research is urgently needed to determine the pathogenic contribution of reperfusion-driven elevated mitochondrial proton leak to the injury, and in particular, the role of cyclophilin D.

### 1.5. Mitochondrial Dysfunction in Diseases Driven by Developmental Failure of Brain and Lungs in Premature Infants

In this section, we discuss two neonatal diseases, chronic lung disease (CLD), also known as bronchopulmonary dysplasia (BPD) and WMI. Both these diseases are associated exclusively with premature birth. Prematurity is known for incomplete pulmonary alveolarization and white matter myelination at birth. With current life-support, both organs continue to mature during extra-uterine life. However, various postnatal stresses may cause the developmental failure of alveolar formation or/and cerebral myelination, which manifests as BPD or/and diffuse WMI. Thus, these diseases are unified by postnatal failure to develop organs’ functional units: alveoli in the lungs and myelinated axons in the brain. Clinically, this manifests as respiratory insufficiency to maintain adequate gas exchange and permanent neurodevelopmental impairment. The mechanisms behind the developmental arrest of white matter myelination or pulmonary alveolarization remain cryptic. Both these diseases are unique to premature infants, and on a cellular level, maturation failure rather than cellular death is a leading pathophysiological process. This fact may be interpreted as a cue for the existence of a common cellular mechanism in the evolution of BPD and WMI. We propose mitochondrial bioenergetic dysfunction as a unifying and fundamental mechanism driving a failure of cellular maturation and differentiation during postnatal organ development.

### 1.6. mPTP in the Pathogenesis of Diffuse WMI of Prematurity

Upon initial description, WMI has been defined as neuropathological and neuroimage-identifiable tissue loss in the periventricular white matter (WM), leading to neurodevelopmental impairment, permanent cognitive and sensorimotor disability. Advances in neonatal care have shifted the neuropathological spectrum of WMI away from a massive degeneration of oligodendrocytes (OLs), their precursors and other cells, known as cystic periventricular leukomalacia (PVL), toward diffuse WMI characterized by global cerebral hypomyelination [64]. Mechanistically, diffuse WMI has been linked to a failure of oligodendrocyte precursor cells (OPCs) and premyelinating oligodendrocytes (pre-OLs) to reach their myelin-producing state [65,66].

Postnatal OL maturation and synthesis of myelin is a highly energy-dependent process [67]. In rats, during the initial 3 weeks of life, when the rate of cerebral growth is the greatest, relative to any other age [68], mitochondrial oxidative phosphorylating capacity in the brain increases by 11-fold, compared to that at birth [69,70]. Energy-dependent biosynthesis in differentiating OLs is very intense, which translates into elevated cerebral metabolic demand during brain development [71]. What type of bioenergetics-affecting clinical stress might be responsible for arresting OLs maturation in premature infants?

In preterm infants, chronic intermittent hypoxemia (IH) is the most common clinical manifestation of cerebral and pulmonary immaturity [72] and has been associated with adverse neurodevelopmental outcomes [73]. In newborn mice, chronic IH-stress during the initial 2 weeks of life reproduced the phenotype of diffuse WMI, which persisted into animal adulthood [40,74]. Importantly, in these studies, IH stress was not associated with cellular death yet triggered mPTP, evidenced by significantly elevated proton leak in mitochondria isolated from the brain or cultured differentiating OLs [40]. This mPTP was sensitive to cyclosporine A, and cyclophilin D deficient mice and cultured OL precursors exhibited normal OLs maturation and cerebral myelination despite IH exposure [40]. Moreover, mitochondria isolated from cyclophilin D knock out differentiating OLs exposed to IH preserved their membrane integrity and were free of mPTP. These data suggest that cyclophilin D-dependent mPTP is activated during sublethal chronic and intermittent hypoxic stress and causes mitochondrial bioenergetic dysfunction evidenced by decreased cerebral ATP content only in wild-type animals [40]. Thus, activation of cyclophilin D-dependent mPTP in response to sublethal IH stress can be viewed as a pathogenic mechanism for OLs maturation failure and development of diffuse WMI in the absence of cellular demise. Of note, in this study, the insight into the upstream cause of activated mitochondrial proton leak revealed a significant Ca^2+^ accumulation in IH-mitochondria. Given cell nonlethal nature of the IH stress, the extent of mitochondrial Ca^2+^ accumulation is probably distinct from the mitochondrial Ca^2+^ accumulation and derangement activating permanent mPTP leading to organelles swelling and cell death in the ischemia-reperfusion and other cell-lethal insults [53,75]. In physiological conditions, mitochondrial membranes may transiently become ion-permeable by activating a low-conductance permeability for Ca^2+^ efflux to mitigate Ca^2+^ overload [51]. This causes temporary mitochondrial depolarization, which in the healthy state is asynchronous among the organelle population, leaving cellular bioenergetics unaffected [76]. IH stress, however, is systemic and simultaneously affects all cells; therefore, a low-conductance mPTP is expected to be synchronized with IH events. Being synchronized, these IH-driven mitochondrial depolarization episodes will negatively affect the organelles’ capacity to produce ATP. Earlier, Bizzozero et al., using various mitochondrial respiration inhibitors in young rats, have shown that formation and compaction of the myelin sheath were highly ATP-dependent [77]. In adult rats, however, the maintenance of the myelin sheath was nearly ATP-independent [67].

Interestingly, while undifferentiating OL precursors (OPC) exhibited resistance to inhibition of mitochondrial complex I, actively differentiating OPC have arrested their maturation in response to mild, sublethal inhibition of mitochondrial complex I [78]. In contrast, protection of complex I with hyperforin preserved mitochondrial ADP-phosphorylating activity and increased expression of the OL maturation-related proteins [79]. These data indicate that normal OL differentiation is sensitive to sublethal mitochondrial dysfunction. An inhibition of complex IV, which resulted in lethal mitochondrial dysfunction, not only arrested OL differentiation but, as expected, caused degeneration of mature OLs and their precursors [80]. Taken together, the above-referenced reports suggest that cell nonlethal mitochondrial dysfunction either driven by transient mPTP or mild inhibition of the respiratory chain arrests normal OL maturation leading to cerebral hypomyelination, the pathological hallmark of diffuse WMI.

### 1.7. Mitochondrial Dysfunction and mPTP in Neonatal BPD

In the US, BPD annually impacts the health of ≈10,000 premature infants. These infants exhibit life-long pulmonary dysfunction [81]. Similar to diffuse WMI, where arrested OL maturation and hypomyelination causes disease, in BPD primary pathological process is the developmental arrest of alveolar formation. At birth, premature infants’ lungs are at the saccular stage of their maturation and not fully functional. As such, premature infants are destined to accomplish their lung development (transition from the saccular stage to the alveolar stage) during extra-uterine life. Among postnatal stresses associated with BPD, exposure to high oxygen concentrations has been considered one of the major triggers of alveolar developmental arrest. At birth, premature infants always require respiratory support. The most common respiratory support mode is prolonged supplemental oxygen therapy, delivered with or without mechanical ventilation. While supplemental oxygen normalizes systemic oxygenation, lungs experience exposure to supra-physiological levels of oxygen. Interestingly, in animals exposed to chronic hyperoxia, accumulation of dysmorphic, swollen mitochondria with abnormal cristae in pulmonary cells has been noted for a long time [82,83].

Similar to brain development, pulmonary development and growth also require substantial energy support. For example, significantly poorer alveolar growth and lung volume at 2 weeks of life have been reported in mice with a slow metabolic rate due to leptin deficiency [84]. Mitochondrial dysfunction, defined by inhibition of oxidative phosphorylation, has been implicated in the mechanisms of the poor proliferation of alveolar epithelium [85]. When BPD-like lung injuries were induced by hyperoxia or mechanical ventilation, the alveolar developmental arrest was strongly associated with the pulmonary mitochondria’s respiratory chain inhibition [86,87]. Moreover, pharmacological inhibition of mitochondrial complex I or partial uncoupling of lung mitochondria fully replicated the phenotype of BPD—arrested alveolarization [86,87]. These preclinical studies suggest that inhibition of mitochondrial bioenergetics in the developing lungs may represent a fundamental alveolar developmental arrest mechanism in premature infants with BPD.

Of note, in the same neonatal mice, exposure to mitochondrial uncoupling agent, 2’4-dinitrophenol, fully recapitulated both alveolar developmental arrest and diffuse axonal hypomyelination [88]. Combined with data on the pathogenic role of cyclophilin D-dependent proton leak in the evolution of diffuse WMI [40], fully reproduced phenotype of diffuse WMI and BPD in the same animals exposed to uncoupler supports the mechanistic role of elevated mitochondrial proton leak in these diseases. However, the role of cyclophilin D-dependent mPTP in hyperoxia-induced lung injury, including neonatal BPD, remains controversial. In neonatal rats, post-treatment with cyclosporine A following 2 weeks of hyperoxic exposure to 60% O_2_ did not improve hyperoxia-induced alveolarization deficit [89]. In mature rodents, pre- and cotreatment with cyclosporine A during hyperoxia protected lungs against acute (72 h) and severe (100% O_2_) hyperoxic insult [90,91]. In the same model, cyclosporine A prevented mitochondrial cytochrome *c* release and swelling and significantly attenuated lung injury [92]. While these reports highlight the pathogenic contribution of cyclophilin D-dependent mPTP activation in hyperoxia-induced acute respiratory distress syndrome-like mature lung injury, Budinger et al. reported no lung protection in cyclophilin D-deficient mice in the same model [93]. Thus, available data on the mechanistic role of cyclophilin D-dependent mPTP activation to mature lung injury or arrested alveolarization in immature lungs driven by oxygen toxicity are limited and conflicting.

With respect to the potential role of mitochondrial ion/proton leak, early induction of uncoupling protein 2 (UCP2) has been reported in pulmonary macrophages during hyperoxia-induced lung injury [94]. Mitochondrial uncoupling was also highlighted in lipopolysaccharide-induced lung injury mechanisms when upregulation of UCP2 exacerbated lung injury and increased animals’ mortality. This was associated with significantly decreased ATP content and moderately reduced mitochondrial membrane potential [95]. In both reports, however, direct induction of mitochondrial uncoupling by UCP2 was not shown. Given that advanced stages of neonatal BPD are characterized by secondary pulmonary hypertension evolving into “cor pulmonale” (right ventricular hypertrophy and diastolic dysfunction), it is worth mentioning the beneficial effect of cyclosporine A on right ventricular mitochondrial integrity and the extent of right ventricular hypertrophy in monocrotaline-induced pulmonary hypertension model [96]. While more data are available on the pathogenic role of mitochondrial dysfunction in pediatric pulmonary hypertension [97]—which evolves in the advanced stages of BPD—data on the pathogenic impact of mitochondrial mPTP and dysfunction to the initial stage of neonatal BPD development are sketchy and limited.

## 2. Conclusions

To a certain extent, the presented literature analysis supports the mechanistic significance of mitochondrial bioenergetic dysfunction driven by excessive proton leak in the pathogenesis of neonatal diseases triggered by cellular death or arrested organs maturation. In neonatal ischemia-reperfusion brain injury due to HI insult [98], the immature organ state is associated with different contribution of cyclophilin D to the damage, compared to that in the mature brain. For example, reported exacerbation of HI brain injury in cyclophilin D knock-out neonatal mice, compared to attenuation of brain injury in the same model in adult counterparts [47]. While significant research efforts in the field are focused on understanding the molecular identity of the mPTP phenomenon, studies on the translational significance, especially in the developing subjects, are scarce and limited to the diseases driven by acute cellular death. There is even less attention to mitochondrial mechanisms of diseases caused by developmental organ failure in premature infants. Given that BPD and diffuse WMI continue to be an overwhelming clinical problem in premature infants, and molecular mechanisms of these diseases are still unknown, our data analysis highlighted an urgent need for more intense involvement of mitochondriologists in this field of medicine.

## Figures and Tables

**Figure 1 cells-10-00569-f001:**
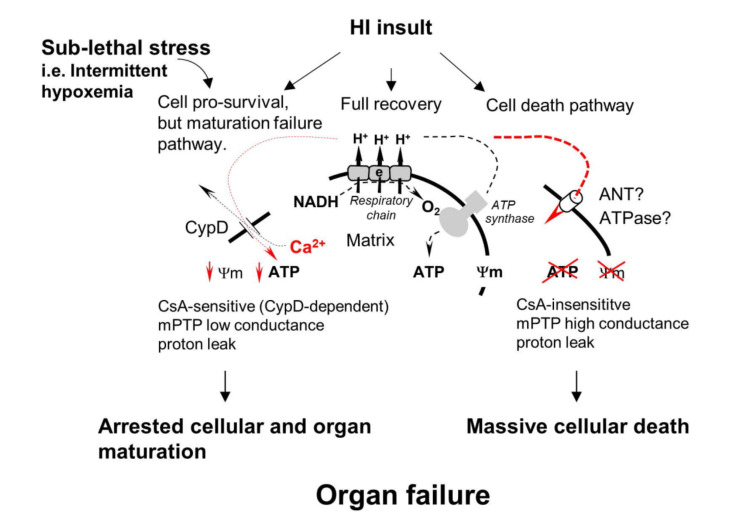
Schematic presentation of hypothetical roles of mPTP in various cellular fates after HI insult in the developing brain: full recovery and survival, death or survival but at the expense of maturation/differentiation failure. Based on current literature, we propose that in the developing brain, CypD-independent (ANT? ATPase? dependent) mPTP activation initiates secondary energy failure and cellular death, while CypD-dependent mitochondrial ion/proton leak promotes cellular viability following cell-lethal HI insult [47]. However, the same CypD-dependent mitochondrial proton/ion leak, via uncoupling of mitochondrial respiration in differentiating oligodendrocytes contributes to their maturation failure to reach myelin-producing stage leading to diffuse white matter hypomyelination [40]. Thus, it is possible that in the developing brain HI injury CypD-independent mPTP contributes to cellular degeneration and CypD-dependent proton leak supports cellular viability but contributes to the post-HI maturation failure of viable/survived cells.

## Data Availability

No new data were created or analyzed in this study. Data sharing is not applicable to this article.

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
