# Peer review of "Mitochondrial Dysfunction and Permeability Transition in Neonatal Brain and Lung Injuries"

_cells, 2021, doi:10.3390/cells10030569_

Round 1

Reviewer 1 Report

In the review "Mitochondrial dysfunction and permeability transition in neonatal brain and lung injuries" (cells-1100527) Ten et al. examine the connection between neonatal hypoxic-ischemic (HI) brain injury, white matter injury (WMI), chronic lung disease/bronchopulmonary dysplasia (CLC/BPD) with respect to mitochondrial dysfunction (primary and secondary energy crisis) and the mitochondrial permeability transition pore (mPTP). The review is well written and it is clear that the authors have made a lot of effort to make the subject, based on many apparently contradictory studies, comprehensive. However, it is questionable to include primary data in a review and, in my opinion, the manuscript should be enriched with illustrative figures. The manuscript is recommended for publication in Cells after the below comments have been considered.

Major comments

Fig. 1. It is odd to include primary experiments (those presented in Fig. 1 and the patch-clamp experiments described at the very end of p. 6 and beginning p. 7 (although the data is not shown) in a review (as the indicated manuscript type). Maybe the authors should consider to publish their experimental data elsewhere. However, this also depends if it is accepted by editorial policy.

The manuscript lacks important illustrations. For example: schematic figures for what happens in mitochondria (as described in section 2); schematics of the primary, secondary energy crises and mPTP in mitochondria; tables of manifestations and mitochondrial dysfunction in HI, WMI and CLC/BPD.

Minor comments

- Introduction, beginning of second paragraph: "... all diseases in newborn infants could be divided into two main categories ...". All diseases in newborns cannot be divided in these two categories, considering that there are many less severe and infections diseases etc, but maybe the developmental disorders can.

- Fig. 1 and the patch-clamp experiments described at the very end of p. 6 and beginning p. 7. It should be mentioned somewhere (text or figure legend) which animal model is being used.

Author Response

The authors are thankful for favorable review and important comments.

Reviewer 1.

  1. 1. It is odd to include primary experiments (those presented in Fig. 1 and the patch-clamp experiments described at the very end of p. 6 and beginning p. 7 (although the data is not shown) in a review (as the indicated manuscript type). Maybe the authors should consider to publish their experimental data elsewhere. However, this also depends if it is accepted by editorial policy.

Response: Thank you, we decided to remove experimental data.

  1. The manuscript lacks important illustrations. For example: schematic figures for what happens in mitochondria (as described in section 2); schematics of the primary, secondary energy crises and mPTP in mitochondria; tables of manifestations and mitochondrial dysfunction in HI, WMI and CLC/BPD. 

Response: We have added the recommended illustration (Fig1)

Minor comments:

  • Introduction, beginning of second paragraph: "... all diseases in newborn infants could be divided into two main categories ...". All diseases in newborns cannot be divided in these two categories, considering that there are many less severe and infections diseases etc, but maybe the developmental disorders can. 

Response: We agree and changed our statement.

  • 1 and the patch-clamp experiments described at the very end of p. 6 and beginning p. 7. It should be mentioned somewhere (text or figure legend) which animal model is being used.

Response: We have removed experimental data and mentioned the model in the text (page 11, line 11-14 from the bottom).

Reviewer 2 Report

This review focuses on the potential mechanistic relevance of mPTP and mitochondrial dysfunction in neonatal diseases caused by acute ischemia-reperfusion or developmental failure of organ maturation during development. The manuscript mostly review old basic data concerning the effects of perinatal HI insults on cerebral cell energy. The review is well written and interesting and deserve publication.

Specific points:

-authors wrote that “….in the developing brain, the acuity of primary energy failure may be greater compared to the mature brain”. However, to get a sustained brain injury in neonates it is necessary both ischemia and hypoxia, differently from adults where only ischemia causes a full injury.

- On page 6 authors present a summary of the events occurring after and HI-insult. A figure that summarize the different mechanisms would be helpful

Page 6 line 3. The sentence that include “in stress conditions of oxidative stress”   need to corrected

Author Response

Reviewer 2.

  • authors wrote that “….in the developing brain, the acuity of primary energy failure may be greater compared to the mature brain”. However, to get a sustained brain injury in neonates it is necessary both ischemia and hypoxia, differently from adults where only ischemia causes a full injury.

Response: The reviewer’s comment relates to an accepted in the neonatology field clinical and experimental term, Hypoxia-Ischemia. This term is used to define a pathophysiology of brain injury caused by birth asphyxia. Birth asphyxia relates to acute collapse of fetal circulation associated with birth activity, therefore the pathophysiology of perinatal brain damage, like it is in the mature brain, is ischemia-reperfusion. Similarly, a pathophysiology of the Levine, Rice-Vannucci model of regional HI brain injury in rodents is ischemia-reperfusion, where ischemia is induced by ligation of the one of the carotid arteries (no ischemia due to a circle of Willis collaterals, but a retrograde circulation in the ligated hemisphere which limits cerebral autoregulation). Then, hypoxic exposure which depresses cardiac output to the extent when retrograde cerebral blood flow collapses in the ligated hemisphere and maintained well in the contralateral side due to preserved cerebral autoregulation. This model produces damage only in the “ligated” hemisphere.

  • On page 6 authors present a summary of the events occurring after and HI-insult. A figure that summarize the different mechanisms would be helpful

Response: Thank you for this suggestion, we have included schematic presentation of discussed events (Fig.1).

  • Page 6 line 3. The sentence that include “in stress conditions of oxidative stress” need to corrected

Response: Corrected

Reviewer 3 Report

The authors have summarized a potential impact of mitochondrial bioenergetic dysfunction in the pathogenesis of neonatal brain and lung injuries associated with premature birth.

However, there are several very important issues that discourage me
from suggesting this manuscript for immediate publication in the Journal.
First of all, the present manuscript is a disproportional presentation of two separate topics: there is just one chapter (see 5.2) dedicated to the analysis of mitochondrial dysfunction and mPTP in neonatal lung disease, whereas the chapters 1-5.1 present data and speculations about the involvement
of mitochondria in perinatal brain injury. Moreover, a very recent review paper coming from the same group has thoroughly summarized possible mechanisms of mitochondrial bioenergetic dysfunction contributing to the
disease-specific pulmonary insufficiency and proposed potential therapeutic targets (see a ref. No 95).
Therefore, I suggest the authors to concentrate on one topic, presumably on mitochondrial dysfunction and mPTP in neonatal brain injuries.

My concern is also related to the overestimation of conclusions derived from a single experiment (see the Fig. 1 and the pages 4-6). It is indicated (see the Fig 1 and a figure legend) that just a single animal (a rat? a mouse?) per group was used. Well, it is obvious that proper methodological details, e.g., isolation of mitochondria, used chemicals and staining protocols are missing. I think, it is not scientifically correct to base the discussion on the preliminary results.

A minor point is related to the Chapter "1. Mitochondria and Perinatal Hypoxia-Ischemia Brain Injury": I could not find a single word mentioning mitochondria.

Author Response

  • First of all, the present manuscript is a disproportional presentation of two separate topics: there is just one chapter (see 5.2) dedicated to the analysis of mitochondrial dysfunction and mPTP in neonatal lung disease, whereas the chapters 1-5.1 present data and speculations about the involvement of mitochondria in perinatal brain injury. Moreover, a very recent review paper coming from the same group has thoroughly summarized possible mechanisms of mitochondrial bioenergetic dysfunction contributing to the disease-specific pulmonary insufficiency and proposed potential therapeutic targets (see a ref. No 95). Therefore, I suggest the authors to concentrate on one topic, presumably on mitochondrial dysfunction and mPTP in neonatal brain injuries.

Response: As per reviewer’s comment, we have removed our preliminary data which created too speculative statements. However, we would like to keep the chapter dealing with neonatal chronic lung disease and mPTP and mitochondrial dysfunction, as mPTP has been discussed here and is absent in the Ref 95.

  • My concern is also related to the overestimation of conclusions derived from a single experiment (see the Fig. 1 and the pages 4-6). It is indicated (see the Fig 1 and a figure legend) that just a single animal (a rat? a mouse?) per group was used. Well, it is obvious that proper methodological details, e.g., isolation of mitochondria, used chemicals and staining protocols are missing. I think, it is not scientifically correct to base the discussion on the preliminary results.

Response: The data mentioned by the reviewer have been removed.

  • A minor point is related to the Chapter "1. Mitochondria and Perinatal Hypoxia-Ischemia Brain Injury": I could not find a single word mentioning mitochondria.

Response: This chapter has been revised and now consists of three parts in which the word “mitochondria” has been used.

Round 2

Reviewer 3 Report

Thank you for addressing my comments.

Author Response

Thank you for your valuable comments